# High Ampacity Carbon Nanotube Materials

**DOI:** 10.3390/nano9030383

**Published:** 2019-03-06

**Authors:** Guillermo Mokry, Javier Pozuelo, Juan J. Vilatela, Javier Sanz, Juan Baselga

**Affiliations:** 1Departamento de Ciencia e Ingeniería de Materiales e Ingeniería Química (IAAB), Universidad Carlos III de Madrid, 28911 Leganés, Madrid, Spain; jbaselga@ing.uc3m.es; 2IMDEA Materials Institute, Eric Kandel 2, Getafe, 28906 Madrid, Spain; juanjose.vilatela@imdea.org; 3Departamento de Ingeniería Eléctrica, Universidad Carlos III de Madrid, 28911 Leganés, Madrid, Spain; jsanz@ing.uc3m.es

**Keywords:** ampacity, carbon-nanotubes, composites, interconnects, electromigration, diffusion, current carrying capacity, miniaturization, electronics

## Abstract

Constant evolution of technology is leading to the improvement of electronical devices. Smaller, lighter, faster, are but a few of the properties that have been constantly improved, but these developments come hand in hand with negative downsides. In the case of miniaturization, this shortcoming is found in the inherent property of conducting materials—the limit of current density they can withstand before failure. This property, known as ampacity, is close to reaching its limits at the current scales of use, and the performances of some conductors such as gold or copper suffer severely from it. The need to find alternative conductors with higher ampacity is, therefore, an urgent need, but at the same time, one which requires simultaneous search for decreased density if it is to succeed in an ever-growing electronical world. The uses of these carbon nanotube-based materials, from airplane lightning strike protection systems to the microchip industry, will be evaluated, failure mechanisms at maximum current densities explained, limitations and difficulties in ampacity measurements with different size ranges evaluated, and future lines of research suggested. This review will therefore provide an in-depth view of the rare properties that make carbon nanotubes and their hybrids unique.

## 1. Introduction 

Electromigration has been occurring for hundreds of years, but it wasn’t until the appearance of integrated circuits in 1966 that the problem became a concern for the industry [1]. Electromigration occurs whenever there is a current flow through a conductor, but the conditions for electromigration to cause failure in a conductor could not be met at the time; for instance, the bulk wires used for conducting current at home would have a maximum current density of about 10^4^ A/cm^2^, but the failure of these cables would occur due to Joule heating and not the electromigration process as such. However, with the irruption of integrated circuits [2] came new problems which no one had predicted, and as the conductor strips were being placed in contact with heat sinks, Joule heating failure mechanism was relegated to a second place, which gave way to the main failure mechanism we still experience today—electromigration. 

The first integrated circuits were made from very thin aluminium lines, a material, which due to its low melting temperature (660 °C) and many small grain boundaries, caused very rapid metal diffusion when enough current densities were passed through. This was the effect of electromigration, although at the time, it was still not completely understood. However, to understand why electromigration is such an important phenomenon in metallic interconnects, it is important to understand the whole process and failure mechanism. When a current is passed through a conductor, the metal ions are exposed to two different forces, which act in opposite direction to each other [3]. One of these forces is due to the electric field and the charge of the metal ions, which causes the positively charged metal ions to be attracted to the negatively charged electrode. However, the shielding of the electrons in the conductor makes this force neglectable for most of the models that have been considered through the years. This, therefore, leaves us with one force, which acts in the same direction as the current flow, and occurs due to the exchange of momentum between the moving electrons and the metal ions. In the case of a perfect lattice, such as those which occur in a homogeneous crystalline structure, there is hardly any exchange in the momentum between the moving electrons and the metallic ions.

However, above 0 K, this perfect lattice cannot occur due to the many defects that can be found [4]; point defects, such as vacancies, caused by the movement of atoms in the lattice, or impurities, due to the material purifications methods which are not 100% effective; planar defects, such as grain boundaries which are caused by two separately growing crystals coming into contact, and causing two different crystallographic directions at the juncture; and linear defects, such as dislocations (both edge and screw type) due to the misalignment of the lattice atoms. Apart from these defects, there is also the fact that above 0 K atomic vibration occurs, and the metal atoms are placed out of their perfect lattice position about 10^13^ times each second [5]. This causes the metal ions to move into the path of the electrons periodically, causing the momentum exchange due to electron scattering, and in turn initiating diffusion of metallic atoms away from that point. This eventually causes a thinning of the conductor, leading to a short-circuit in places such as grain boundaries, or vacancy defects, where there is free space available for the moving atoms. At the same time, in grain boundaries, there is a change in the symmetry that characterizes the bulk crystal lattice, making the collisions between the moving electrons and the metallic atoms more frequent. Grain boundaries are also known for having weaker bonds than the rest of the crystal, so all these factors makes them preferential sites for electromigration.

Although metal behaviour is the one being referred to constantly, it is important to mention that electromigration has also been seen to occur in semiconductors that have been heavily doped [6]. In n-type semiconductors, where electrons are the main charge carriers, the diffusion occurs in the same direction as the current flow, whereas in p-type semiconductors, where the charge is being carried mostly by holes, the diffusion phenomenon occurs in the opposite direction to the flow.

Interconnects, as is possible to deduce from the explanation above, being the place where the current pathways change from being macroscopic to being usually smaller, is one of the first places for failure to happen. The conductor dimensions usually change at this point, making the current density to be much higher once it enters the smaller circuit’s cross section areas. If the power required for these electronical devices is always increasing, and the dimensions of the conductors are constantly being decreased, it is easy to figure out that a limit will be soon reached and will not be long until electromigration becomes the limiting factor. As well as miniaturization, there has been an increased research interest in finding materials that can withstand huge current densities in the aeronautical industry. Current lightning strike protection systems are made of copper expanded foils, and if they are to withstand the 200 kA that a lightning can produce [7], they must have a thick cross section area. If a different material, such as a carbon nanotube-copper composite, was to be used in modern trains or commercial aircrafts, it would be possible to reduce the weight by one third, which would mean more than 25,000 tons of fuel saved and a reduction of CO_2_ emissions of around 78,000 tons per year [8]. According to the international roadmap for devices and systems [9], it is of utmost importance to find non-Cu solutions to cope with the increased electromigration risk due to the decreased volume of metal and the ramping of the current densities used, and one of the proposed solutions is the use of carbonaceous materials, such as carbon nanotubes, which will be the common factor of this review, being the final aim of this work to explain with ease a complicated phenomenon that is, and will be, a barrier for technological advances.

## 2. Calculating Ampacity Theoretically 

In 1961, Huntington and Grone [10] reported the electromigration phenomenon for the first time by using light scratches on the surface of a gold wire, to which they applied a current density of 10 kA for days and observed how these markers moved. This was evidence enough to believe there was a driving force which had to do with the current density and the self-diffusion constant. This gave birth, to the famous Huntington and Grone’s equation (Equation (1) below) which is used today:(1)$$J=\frac{ND}{kT}Z^{*}e\rho j=\frac{ND_{0}}{kT}\exp\left(-\frac{\phi }{k_{b}T}\right)Z^{*}e\rho j$$
where *J* is the particle flux, *N* is the particle density, *D* the diffusion coefficient, $k_{b}$ and *T* Boltzmann constant and temperature, respectively. *Z** is the effective valence also known as the effective charge, *e* the charge of an electron, ρ resistivity and *j* current density. Lastly, *ϕ* is the activation energy of diffusion and *D*_0_ is a factor that expresses the frequency of the collisions. From this equation, it is possible to deduce that the sign of the effective valence will give the direction of movement for the atomic migration. If migration occurs for example in the same direction as the electron flow, then the effective valence value will be negative. From Equation (1), it is possible to observe that the mass flux caused by electromigration is directly proportional to the current density, the diffusion coefficient and the concentration of diffusing atoms. 

This equation was used for years, until 1968, when James R. Black [11] found that the electromigration driving force was not proportional to the current density, but to the square of current density. In his opinion, the first models that had appeared for electromigration were not sufficiently exact, as they had used a very narrow distribution of current densities, so he tried to replicate the measurements with a wider range of current densities and soon realized that the mean time for failure of the conductors depended inversely on the square of the current density. This gave way, for Black’s law as seen in Equation (2), which is the one used up until today for electromigration measurements:(2)$$\frac{1}{MTF}=Aj^{2}\exp\left(\frac{\phi }{k_{b}T}\right)$$
where *MTF* is the mean time to failure; *A*, the cross-section area of the film; *j*, the current density; *ϕ*, the activation energy; $k_{b}$ Boltzmann constant and *T* the film temperature. In this same study, it was also observed that the mass transport during electromigration occurred preferentially through the grain boundaries and the surface of the conductor and could be measured through their activation energies: in the case of an aluminium conductor with fine grains, the activation energies that were calculated from the equation above were in the order of 0.48 eV, whilst in well-ordered, large-grained aluminium the activation energy raised to 0.84 eV. This increase in the activation energy was quickly attributed to the higher energy surface and bulk diffusion pathways, which the atoms had to follow when the low energy grain boundary diffusion pathways had been severely reduced. It was also noticed, that above 275 °C, the predominant diffusion was through the lattice and therefore grain and surface diffusions were not important. However, things were not so simple, and it was soon realized that both Black’s law and Huntington’s law were right, and it all depended on the type of failure that occurred, if it was nucleation or growth dominated, which depends strongly on the method used to construct the line. Most aluminium lines for instance, are often placed on top of a refractory metal known as shunt. Through these shunt layers, the electric current can bypass the void formed in the conducting line when electromigration occurs, and in these cases the failure mechanism is growth dominated, and failure occurs once the void is too big and breaks connection, and in this case a 1/*j* kinetics must be used. However, if there is no shunt layer, then the failure is nucleation and migration dominated and follows Black’s law instead. On some occasions, it is even possible to have both nucleation and growth kinetics at the same time, and a power on the current density cannot be used to explain it. These two mechanisms are represented on Figure 1.

There are several diffusion paths available through a conductor, namely lattice, grain boundary, material interfaces and dislocation cores. Lattice diffusion has the highest activation energy [12], which means that the mass transport through this pathway will be the slowest, whilst grain boundaries and interfaces have lower activation energies, which means that mass transport will be predominately through these paths. On the lowest activation energy level, we have surface diffusion. However, this diffusion mechanism is blocked whenever there is a passivation layer on the surface of the conducting line, so it will not be available for metals such as aluminium or copper that form passivation layers, whenever exposed to air. Therefore, to account for the effective diffusion (total diffusion occurring), it is necessary to add all of these individual diffusion pathways with their corresponding fraction of atoms that diffuse through them. This can be seen in Equation (3) below:(3)$$D_{eff}=D_{lattice}+f_{gb}D_{gb}+f_{i}D_{i}+f_{c}D_{c}$$
where *D_eff_* is the effective diffusion coefficient; *D_lattice_*, *D_gb_*, *D_i_* and *D_c_* the diffusion coefficients for lattice, grain boundary, material interface and dislocation cores, respectively; *f_gb_*, *f_i_* and *f_c_*, are the fraction of atoms diffusing through grain boundary, material interface and material cores respectively. For a dual damascene interconnect, the effective diffusivity according to bibliography [13] can be expressed in terms of thickness of grain boundaries ($\delta_{gb}$) and interfaces ($\delta_{i}$), average grain diameter (*d*), dislocation density ($\rho_{c}$), cross section area of the dislocation core ($a_{c}$), line width (*w*) and line height (*h*).
(4)$$D_{eff}=D_{lattice}+\delta_{gb}\left(\frac{w-d}{wd}\right)D_{gb}+\delta_{i}\left(\frac{w+h}{wh}\right)D_{i}+\rho_{c}a_{c}D_{c}$$

The diffusion coefficients are expressed by Arrhenius law. From Equation (4), one can observe that the effective diffusivity depends strongly on factors such as the microstructure, temperature and quality of the interface between the conducting layer and other layers. At higher temperatures, the diffusion rate increases whilst the opposite happens with lower temperatures. Also, if there is a temperature gradient along the line there will be regions with different diffusion rates resulting in hillocks and void formation indistinctly. Thus, temperature has a huge effect on mass diffusion during electromigration. 

On the other hand, microstructure is another factor that greatly affects mass transport. A film with a uniform size of grain will have a network of grain boundaries, which will meet at triple points [12]. A representation of this triple points can be observed in Figure 2.

At this triple points, crystallographic directions change from one grain to the next, so it is easier to find divergences in mass transport at these points. Depending on the type of triple point found, it is possible that one grain boundary leads to two other grain boundaries, in which case there will be more mass leaving the triple point than entering, so a void will form (causing eventually an open circuit if no shunt layer is present), or two grain boundaries joining with a single grain boundary, leading to a mass accumulation and a hillock formation (causing a short circuit between adjacent lines). Following this idea, smaller grain sized metals will lead to the formation of more grain boundaries (and therefore triple points) than bigger grained metals.

From Equation (4), it is also possible to observe some design rules that should be taken into consideration. As illustrated in Figure 3, if the line width is larger than the grain size, it will contribute to the overall electromigration by creating pathways of grain boundaries throughout the line (polycrystalline line), but oppositely, interfacial diffusion will become the preferred diffusion pathway when the linewidth is smaller than the average grain size, as there will be no continuous path of grain boundaries throughout the line (bamboo-like structure).

This fact is specially observed in contacts and vias, as there is an impossibility of the diffusing metallic atoms to cross the materials used in them, and therefore there are voids and hillocks formed at the contact. An example of hillocks formation when the maximum ampacity is reached can be observed in Figure 4. This image was obtained during an experiment with a graphene-copper composite, which was synthesized by our research group.

Electromigration, however, does not only depend on the current density as it was shown by Blech [14] and depends also on a critical length of the line being used. Blech effect consists of three separate concepts—Blech product, blech length and blech condition, all of them are related to the stress gradients which opposes the mass flow during electromigration. As stress gradients build up in the lines to oppose the electromigration force, hillocks and voids help relieve these stresses (also known as back stress). However, after some time, the formation of a void might cause an open circuit if a shunt layer is not present to bypass this point, and a hillock might cause two separate lines to come into contact, causing a short circuit. This stress factor was therefore incorporated into Equation (1) to account for this new observed parameter and this way, Equation (5) was obtained:
(5)$$J=\frac{ND}{kT}(Z^{*}epj-\Omega \frac{\Delta \sigma_{nn}}{\Delta \sigma_{l}})$$
where Ω is the atomic volume, $\sigma_{nn}$ the stress normal to the grain boundaries and *l* is the distance along the line. From this equation comes the first of Blech concepts, the “Blech condition“ as shown in Equation (6), which when satisfied, the overall electromigration driving force will be 0 and therefore no electromigration would occur.
(6)$$Z^{*}epj=\Omega \frac{\Delta \sigma_{nn}}{\Delta \sigma_{l}}$$

If one integrates the Blech condition over the length of the line, one would be able to observe that the stress varies linearly along the line when the backflow flux equals the electromigration flux. Given that there is a limit of stress that a conductor can withstand before failure ($\sigma_{th})$, a critical product for electromigration failure could be stated as:(7)$${\left(jl\right)}_{critical}=\frac{\Delta \sigma_{nn}\Omega }{Z^{*}e\rho }$$
where $\Delta \sigma_{nn}=\sigma_{th}-\sigma_{0}$, being $\sigma_{0}$ the stress at length 0. This is known as “Blech product”. From this equation, for a given current density *j*, one would be able to determine a critical length at which no electromigration would be able to occur. This is the third of Blech conditions, known as “Blech length”, and was observed whenever the line length reached a critical value, as in these cases, electromigration could be avoided. Blech length equation can be observed in Equation (8) below: (8)$$l_{B}=\Omega \frac{\Delta \sigma_{nn}}{\left|Z^{*}\right|e\rho j}$$

## 3. Determining Ampacity Experimentally 

The review has been centred on explaining the electromigration process; however, no importance has yet been given to the term “ampacity”. Ampacity is the maximum current density that a material can withstand before failure occurs. Electromigration, as it has been explained before, starts the moment the first electron passes through. However, there must be enough “wind force” [15] to create momentum exchange to make the lattice atoms move. The term “ampacity” therefore refers to the point at which movement of lattice atoms occur, and therefore resistivity begins to increase. This was defined perfectly by Subramaniam [16] as “the maximum current density at which the resistivity remains constant”. Ampacity, as it has been explained above, could be theoretically calculated using the above formulas, but for most of the new emerging materials, some parameters such as activation energies would be unknown. Therefore, there is an emerging necessity to develop methods that would allow to measure ampacity at any size scale.

Generally, with microwires, the way to measure ampacity is straightforward. However, a 4-point-probe measuring method is always recommended, as this would remove any error that could arise from the cabling being used. By running a current-voltage cycle, one would be able to determine (by using Ohm’s law) the resistance of the material at each different current. This would allow the user to determine resistivities for each current measured during the test. The current at which the value for this resistivity increases would be the one to use for calculating the current density, by dividing by the cross section of the material. 

This kind of measurement has been done in the micrometre and sub-micrometre scale before, and even simultaneous characterization of the material can be done by using techniques such as STM or AFM, which allow for the live visualization of how the morphology changes with current [17].

A good guidance for determining the kind of diffusion that is occurring (lattice, surface or grain diffusion) is that one would require to know the activation energy of the composite material. Higher activation energies would suggest that the diffusion pathways chosen by atoms for electromigration would be less energetically favourable such as lattice diffusion, whilst smaller values would suggest grain diffusion. Determining activation energies can be done with any conducting material, by plotting an Arrhenius plot, in which the slope would correspond to the activation energy.

However, it is important to notice that size would be a big limitation when choosing the right power unit, as bigger cross sections would require more current to be able to reach the ampacity point. If we take bibliographical values for copper ampacity [18], we would need to apply a current of approximately 75 KA to reach the failure point of a line with a cross section of 400 micrometres. One way to avoid these huge currents would be to diminish the cross section of the desired material, but then other phenomena could come into play, such as changes in crystalline structure (from polycrystalline to bamboo type for instance) or going below Blech length, changing thus the diffusion process, which would result in an erroneous ampacity measurement. On the other hand, if instead of using a line, there was a requirement for measuring the ampacity of a bigger 2D material, the difficulty would be even higher. Our group is currently working on optimizing a method we have developed for measuring ampacity in thicker layers of material. 

The system, which can be observed in Figure 5, consists on creating a laboratory-made “lightning strike”, by using a set-up of 12 capacitors in parallel. The laminated sample is placed on a wooden surface, and a collector ring (made of thick copper) is placed on the outside of the sample to prevent any possible over-currents. A sharp copper electrode is placed above the surface of the sample and a drop of salt water is used to create a conducting channel between the tip and the sample. A Rogowski probe is used to measure the current that passes through the electrode once the safety switch is turned on. As soon as the circuit is closed, the current will cause copper to migrate away from the contact zone, creating a small gap that will become bigger together with the radial propagation of the current. The final size of the gap would give the user the cross section of material that the current could not move, therefore giving a value of maximum current density. It has been observed, in materials with a known ampacity, that there is a relation between the damage zone and the maximum current density of the material. However, optimizations must be done, as there is energy being lost through Joule heating, sound, light and small projection of material, giving variable errors in the measurements. Even though measurements are still being carried out, this is a good starting point to be able to measure in the near future the ampacity of materials with other sizes and shapes, other than microwires.

## 4. CNT as High Ampacity Material

### 4.1. Pure CNTs as High Ampacity Materials

The first interconnects, as it was explained at the beginning, were fabricated with aluminium, but by the end of the 1960s Blech exposed the idea that aluminium ICs were failing due to electromigration [19]. Black [11], several years later, systematically studied the phenomenon varying several parameters and came up with an equation by which, theoretically, one could construct aluminium lines with infinite lifetime, by taking into consideration current densities, conductor temperatures and conductor cross sectional areas. However, it was soon discovered that these infinite lifetime lines could not really avoid electromigration due to grain structures and other defects, and soon many articles related to this electromigration phenomenon topic appeared [19,20,21], which explained the relation that was observed between the aggregates that were being formed at exactly different grain sizes transition points. Exactly two-thirds of the defects that were being found in the aluminium lines were at these transition points. This can be easily understood by looking at the activation energies for the different kinds of diffusion in aluminium; bulk diffusion had an activation energy of 1.2 eV whilst surface diffusion was 0.8 eV. On the other hand, grain boundary diffusion had an activation energy of 0.7 eV, which meant that this diffusion path was the most favourable energetically speaking [22]. 

Soon, copper was found to be a good replacement for aluminium, both by doping aluminium with copper, and by using pure copper as the line material. Copper, in comparison to aluminium has different diffusion activation energies—bulk, grain-boundary and surface diffusion were in the order of highest to lowest activation energies, making surface diffusion the preferred mechanism in copper (2.3, 1.2 and 0.8 eV, respectively) [22]. However, copper in contact with air rapidly oxidizes, which prevents diffusion through the surface, and therefore grain boundary becomes the primary diffusion pathway. Copper, however, is only a temporary measure, as miniaturization is pushing the cross-section areas of the copper lines to new limits, which exceeds the ampacity values of the metal. As a point to consider, the technology used for the fabrication of copper, namely damascene technology [14], has also been seen to affect the quality of the copper lines produced, as due to the high temperatures used, diffusion of copper ions into the neighbouring silicon or silicon dioxide has been observed [23]. Therefore, further steps are necessary to prevent this diffusion problem, which would mean an increase in the total costs for the process. 

A new field has emerged over the last few years, by which ampacity can be improved by using carbon nanotubes. Carbon nanotubes are one of the several structures that elemental carbon can form in its sp2 hybridization. In 1991, Iijima observed this kind of tubular structure and gave them the name by which we know them today [24], and since then many new techniques have been developed to produce multi-walled carbon nanotubes [25,26,27], single walled carbon nanotubes [28,29,30], carbon nanotube yarns [31,32,33], films [34,35] and composites [16,36,37,38,39,40,41], to name a few. Some examples of the structures that can be obtained with carbon nanotubes are shown in Figure 6. Even though this review is centred around electrical properties of the nanotubes, it is important to mention that there is also a close relation between crystal structure the mechanical properties of these tubes, which must be understood if they are to be used effectively on their own or as part of a composite material. Salvetat et al. [42] demonstrated that the orientation of the carbon nanotubes had a great impact on mechanical properties. Highly ordered nanotubes, produced by arc discharge method, were seen to possess high elastic moduli of around 810 GPa. However, other nanotubes, which had been produced by catalytic methods, showed a steep decrease of these values and were close to 50 GPa. The explanation was that the elastic behaviour of disordered nanotubes could involve shear deformation, which is much more sensitive to any defects or dislocations present that the highly ordered nanotubes. These highly ordered nanotubes showed no change in elastic modulus, even when subjected to thermal anneals to reduce point defects. This led the author to the conclusion, that in highly ordered carbon nanotubes, point defects have no effect on mechanical properties, which makes these kinds of nanotubes the best option to use for industrial purposes, as we have been mentioning throughout the review. It is important to mention that in such ordered cases, the diameter of the nanotubes seem to have no effect on the elastic modulus. Other articles, such as the one by Yamamoto et al. [43], also showed that the elastic modulus could be enhanced by using thermal annealing methods. MWCNTs were annealed at 1800 °C, 2200 °C and 2600 °C and these showed improved strengths by a factor of 5.4, 5.1 and 15.6, and their elastic moduli by a factor of 5.9, 13.2 and 18.9, respectively. Their conclusion was that by annealing at such high temperatures, the degree of orientation of the 002 graphitic planes was improved, and the number of defect concentrations was severely reduced. Also, the undulated and disordered structures changed to a nearly perfect cylindrical structure when annealed at such temperatures, which changed the kind of fracture from clean break type, to sword-in-sheath failure. These factors, as well as affecting the mechanical properties, also affect the electrical properties, as perfect lattices improve conductivity greatly as there is no defect to oppose the flow of electrons through the nanotube structure. As has been explained in the sections above, defects are usually the points at which electromigration preferentially occurs. However, it is a different, and equally important property of these nanotubes that will require our attention in this review—their resistance to high current densities, or in other words, their ampacity.

Carbon nanotubes exhibit a high tolerance to electromigration, due to the strongly bonded system that occurs naturally between carbon atoms [44]. To give an exact number, the ampacity of carbon nanotubes has been found to be close to 1 × 10^9^ A/cm^2^, three order of magnitude lower than those of copper or gold (1 × 10^6^ A/cm^2^ approximately) [45]. Not less important is the fact that the thermal conductivity of carbon nanotubes exceeds that of copper by a factor of 15, which makes them the perfect option for dissipating heat from sensitive areas such as interconnects [46]. However, these conductivities may vary greatly with the structure of the carbon nanotube, as was observed in 1996 by Ebbesen et al. [47], when they discovered that even carbon nanotubes that had been prepared at the same time and with the same method (by arc discharge method) exhibited different electrical properties from one tube to the next. This came as a big surprise, as the tubes even behaved differently when subjected to temperature variations. Some tubes had a metallic temperature dependence whilst others were clearly semiconducting, and it was concluded that helicity and interlayer interactions played an important role in these electrical properties. Carbon nanotubes present a type of conduction which can only be compared to superconductors, as it has been proved that electrons move through them by ballistic transport, meaning that there is no resistance to the flow of electrons [48]. Ballistic transport occurs when the length of a conductor is smaller than the electron means free path (approximately 40nm), and in these cases each conducting channel would contribute to the total conductance with a unit known as conductance quantum and usually denoted by G_0_ (G_0_ = 2e^2^/h, where e, is the charge of an electron and h, Plank’s constant) [48]. This has been proved in single walled carbon nanotubes easily by standard measuring methods (2 and 4 terminal sensing) and shows that as single walled carbon nanotubes have two bands crossing the Fermi level, the total conductance would be given as expected in a metallic nanotube, by 2G_0_. However, some measurements in MWCNTs have shown that the conductance of SWCNTs cannot be applied to each shell of a MWCNT individually, as it has been observed with a conductance below 2G_0_ in most cases. The cause for these low conductance quantum measurements can be found in the current probing methods, as these will only consider the outer shell of the MWCNTs and not every layer [49,50], and it is well documented that the interaction between successive walls of the tube reduces the total conductance [51].

One would say that, knowing this conductance quantum properties, SWCNTs would be the best possible solution for interconnects. However, due to the synthesis methods used to produce these nanotubes, their chirality changes and with it the electrical properties they exhibit [47,52], making it possible to have either metallic or semiconducting type single-walled nanotubes. As it is easy to understand, it is the former kind that is of interest to improve the current metallic interconnects [53,54,55,56,57]. On the other hand, multi-walled carbon nanotubes usually show a metallic behaviour with most of the synthesis methods that are used [58]. It is interesting to notice that multi-walled carbon nanotubes do not fail at high current densities via electromigration, as has been explained before, but fail in a series of sharp steps associated with the destruction of individual nanotube shells [45], as has been observed via AFM, showing how multi-walled carbon nanotubes changed their diameter with increased current densities. 

However, carbon nanotube technology has a few problems when researchers try to use these nanotubes on their own as interconnects. One of the first issues found when producing single or multi-walled carbon nanotubes are the high temperatures required for the synthesis and deposition on the substrate, as it affects the transistors and doping profiles. Many efforts are directed towards process engineering in order to try and reduce the temperatures being used for this synthesis. Advances have been achieved recently and temperatures closer to 350 °C have been proven to be enough to this end [59] by using cobalt as the catalyst and obtaining good uniformity with no apparent yield loss compared to higher temperature synthesis. However, some difficulties must be overcome if these CNTs are to be used, as resistivity values of the nanotubes produced at low temperatures are still above those exhibited by high temperature synthesis.

More difficulties have been observed when using carbon nanotubes as interconnects due to the problems that occur when contacting them to other materials, usually caused by the fermi energy differences between both and to the metal-carbon nanotube interface chemistry [60]. This causes very high contact resistance (usually greater than 6.5 KΩ for SWCNTs [61]), which severely reduces their lifetime. Carrier tunnelling across the interface of nanotubes and metals due to the very different work functions between the two is the main cause of these high contact resistances. However, other causes such as defects (which include impurities) on either the nanocarbon or the metal would also cause higher contact resistance. To this end, constant research in this field is being carried out, looking to improve the contact resistance, as no matter how good the conductivity of the nanotubes or the metal, this will limit the overall performance for interconnects [60].

Another consideration for making carbon nanotubes a viable material for interconnects is that the nanotubes must have a metallic behavior, as has been mentioned above. Although usually to this end, and due to ease of production, multi-walled carbon nanotubes are used, it has been found that although contact resistance is indeed reduced with metallic type nanotubes, resistivity remains two orders of magnitude above the one of copper [46], as was mentioned above, due to the interaction of each of the MWCNT walls. To reduce this resistivity, there have been some recent advances whereby using titanium particles as catalysts, the metal remains at the tip of the nanotube after the growth process and brings all the layers into contact. This causes a big decrease in their resistance, making this process interesting for research in the near future [46]. 

Finally, but not less important, longer lengths than the ones exhibited by a single nanotube must be connected, therefore needing not a single nanotube that is far too short, but a bundle of nanotubes instead. As has been mentioned, controlling chirality in carbon nanotubes is very difficult; therefore, in a bundle, it is possible to find both metallic and semiconducting nanotubes, which would not contribute to the overall current conduction. This, together with the fact that in between the nanotubes there will also be a contact resistance makes longer CNT bundles less efficient than shorter CNTs if a higher ampacity is searched for. However, and even with these problems, there are some authors who claim to have developed high-ampacity power cables, made of closely packed and aligned carbon nanotubes [62], with values of ampacity close to 1 × 10^5^ A/cm^2^. Other recent results have shown that by using a radial densification process, it is possible to obtain cables with a diameter up to 1 cm in size [63]. A densified core of CNTs is first produced, and then successive layers of CNTs are wrapped and densified on top. This method produced a conducting cable that can withstand much-higher Joule heating temperatures than normal copper cables, which would make it an outstanding material for high current cables in air. 

This result shows how promising the field of carbon nanotubes as high ampacity material is, and surely, soon, the drawbacks that are being encountered will be overcome, granting technology with a cheap and effective substitute for copper. However, advances are yet to reach this point, and until then, and even though carbon nanotubes by themselves are interesting as has been explained above, it is their use as composites that has garnered great attention in recent years. 

### 4.2. CNT Composites as High Ampacity Materials

The possibility of using the outstanding conductivity values of metals, together with the great ampacity behaviour of carbonaceous materials, might be the solution to cope with the future needs in miniaturized electronics. However, there are some issues when synthesizing carbon nanotubes—metal composites due to the poor adhesion that metals have shown with CNTs [64]. The very strong covalent bonds that carbon atoms show towards each other prevent the need to bond to other kinds of atoms, such as the ones in outside metals. However, treating these carbonaceous materials with a pre-oxidation, acid or a heating stage after producing them have shown improvements in adhesion [37]. Metals surface tension plays an important role when trying to wet the carbon nanotubes, and only liquids with surface tensions below 100–200 mN/m can wet this carbonaceous material [65].

For achieving these composite materials, several techniques have been used [40]; the technique that is most utilized is powder metallurgy [66,67,68], followed closely by electrodeposition and electroless deposition (being nickel and copper the two metals which have been most used) [69,70,71,72,73,74,75,76]. An example of the kind of structures that can be obtained by using electrodeposition on carbon nanotube can be observed in Figure 7, where the author synthesized a CNT-Cu structure by this kind of electrochemical growth. For metals with a low melting temperature, a viable option is melting and solidifying them in the presence of carbon nanotubes [77,78,79]. In addition to processes for synthesizing these composites, there are many metals that have been used in order to improve different properties. However, and due to the scope of this review, we will be centering our attention on the composites that pursue the improvement of ampacity as its end goal. These composites are composed of carbon nanotubes and copper, which is the metal that has proven to exceed any others in conductivity, and the one that is used up until today for interconnects and most current conductors. 

Although CNT-Copper composites were synthesized several times with success [75,80,81,82,83,84], it was not until 2008 [85] when the idea of how this composite could behave under high current densities was presented. In this paper, a low-density CNT matrix was synthesized by CVD, and the voids in between tubes were filled up with copper by electrodeposition. After the annealing of the composite material, it was observed that the average growth rate of voids caused by electromigration was four times smaller than the one observed for pure copper. Several years later, in 2013 [16], Subramaniam et al. went further and stated that their copper-SWCNT composite showed a 100-fold increase in the ampacity when compared to pure copper and showing no decrease in conductivity. Their composite material was made by depositing vertically aligned carbon nanotubes (densified by liquid densification technique), which when subjected to shear force became horizontally oriented. Then, in a two-stage growth process, organic electrodeposition followed by a second phase of aqueous electrodeposition, SWCNT-copper composite was achieved. It is very important to explain that the organic phase was used to wet the CNTs so that the copper would be able to adhere and nucleate on the nanotubes. A second important lesson was obtained from this article—that the rate-limiting step had to be the copper nucleation and not ion diffusion during electrodeposition. To this end, a slow deposition rate had to be used in order to achieve ion diffusion towards the interior of the CNT structure and prevent deposition like this on the outside only (as that would have made a coating of copper and not a proper composite material). As was stated, CNTs seemed to prevent low energy diffusion pathways or, in other words, diffusion through surface and grain boundaries, forcing diffusion through the bulk of the lattice. This was proved by plotting experimental results, and using Black’s law to calculate the slope, giving a value of diffusion which was the same as the activation energy of lattice diffusion in pure copper (2.03 eV). 

This same composite material was researched in other articles [86], showing as well as the mentioned outstanding properties, a metal like thermal conductivity (395 W m^−1^ K^−1^) and a silicon-like thermal expansion. This expansion coefficient is of utmost important, as this is one of the main reasons for overheating of semiconductor-based electronics. At the interface of the heat sink (made of metals such as copper or aluminium) and the substrate (silicon) is a huge mismatch in thermal expansion coefficients (approximately 300% mismatch), which causes failure at these points due to delamination [87]. In order to prevent this from occurring, similar thermal expansion coefficients must be present for both the heat sink and the conducting heat sink, which can be obtained (only a 10% mismatch in comparison) by using a copper-carbon nanotube composite material, making it once again the best option for microelectronics. Other properties, such as outstanding performance to nanoindentation when compared to copper, have also been achieved for these composites [88].

Several electrical models have been made [89,90] to analyse how this composite material would behave in real chips. The obtained results show that with increasing length, CNT-Copper composite can outperform both pure copper and carbon nanotubes. 

All these results show that in micro-scale carbon nanotube-copper composites are probably unrivalled. However, ampacity does not only occur at this scale, and therefore macroscopic properties of this composite must also be studied. For instance, if new materials are to be found for lightning strike protection to substitute the current expanded copper foil technology used in aircrafts, other scales of production must be studied, as CVD would not be able to cope with industrial scaling of the mentioned technology. Other techniques have appeared recently, which allows a much bigger production of CNT ropes and yarns than the methods mentioned above. One of those methods starts from the same point as the ones above, and CVD must be used to grow a vertically aligned forest of CNTs [31,91]. From this point, one is able, by a pulling method, to create a fibre of oriented CNTs. However, this method has its limitations in industrialization, as again chemical vapor deposition would be the limiting scaling factor. However, other options to create this same CNT films by industrial methods have recently appeared [92,93,94,95]. These films as well can be condensed by using any alcohol to create a densified CNT fibre, an outstanding start for industrial CNT-copper composites.

At the microscale, several properties such as the size of grain in comparison to the line diameter, or the total conducting path length, can cause drastic changes in the ampacity values. So as to speak, the articles that have been named above have found the optimal parameters for outstanding ampacity values in their composite material. However, this optimization had to be done at macroscopic levels too, where grain size would cause a much-smaller effect on the overall electromigration values when compared to the much-bigger diameter of the conducting path. Some other effects, related with the starting CNT matrix, can also be a drawback when synthesizing the final copper-CNT composite material by the electrodeposition methods. As mentioned in several papers, with growing sizes of the CNT lines being used for electrodeposition comes a decrease in the distribution of charge when undergoing electrodeposition [96]. The area closest to the electrodes would show a higher copper deposition rate than the areas further from the electrode. It was demonstrated that only CNT cables of maximum 10 cm in length could be used to produce uniformly distributed CNT-Copper composites. Further insight must be given to this issue if a continuous production of CNT-Cu composite is intended, although several articles have moved towards this direction [97]. The ampacity value of this macroscopic composite material, however, showed a severe decrease when compared to the microscopic version of the same material. The current carrying capacity at this size level showed only slight improvement, when compared to pure copper. Other articles have mentioned that ampacity could even be slightly reduced by the addition of carbon nanotubes, probably due to the low average conductivity of the carbon nanotubes used [98]. Either way, one thing is for sure; large and homogeneous conductivity along the transport direction in interconnects is of vital importance to avoid electromigration [99]. Adding an intermediate layer of Nickel has proved beneficial for improving the adhesion and deposition of copper, which in term increases the effective strength of the composite, quality of the deposited layer, conductivity and ultimately ampacity [100]. A short summary of some of these studies is presented in Table 1, where, for comparative reasons, pure metals, pure carbonaceous materials and metal-carbonaceous composite materials are placed together to demonstrate how composite materials seem to be the way forward in improving most electrical properties.

## 5. Conclusions

In this review, we have tried to present an insight into this emerging field of carbon nanotubes and carbon nanotube-composites as high ampacity materials. Alternatives to the current copper or aluminium components must be found before the limit of current carrying capacity is reached. That limit seems to be very close according to analysed trends, so any field that seems promising must be studied. One of them, carbon nanotubes, seems to be the most promising of them all, and this is shown by the many bibliographical references to both CNTs and ampacity that can be found on the web, in books and research articles. CNT, together with copper, seems to be the most promising composite material up to date. However, there are still many areas open to discussion, as some articles have found contradicting results on the quantification of the ampacity of such composites. In most cases, a significant improvement has been observed, whilst in a few cases, the composite material exhibited lower ampacities than those shown initially by copper on its own. Special attention should be given to the fact that this technology is mostly centred on the micro-scale; however, there is also a wide field of interest on macroscopic-sized composite materials, for their use in aeronautical or rail industries. Research on this size range is scarce, and therefore offers great margin for improvement. Ultimately, this will open the doors for future technologies, where carbon-based conductors will replace metallic conductors, as we know them.

## Figures and Tables

**Figure 1 nanomaterials-09-00383-f001:**
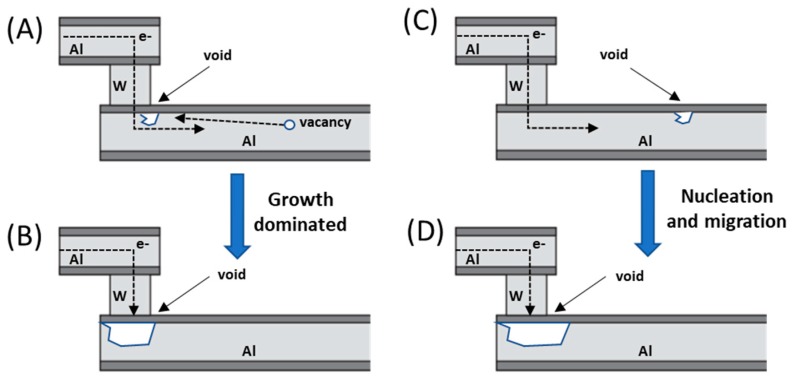
Two different mechanisms of void formation in copper during electromigration stress for (**A**) and (**B**) which is due to growth dominated failure, and (**C**) and (**D**) which corresponds to nucleation and migration.

**Figure 2 nanomaterials-09-00383-f002:**
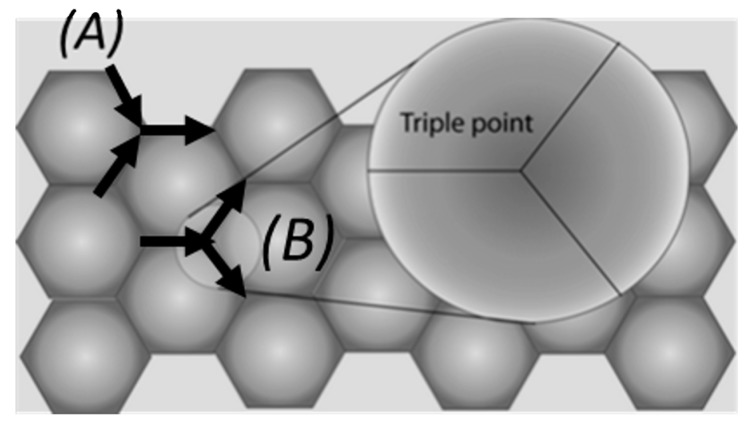
A triple point region is the junction where crystallographic directions change from one grain to the next. These points are the usual spots where regions of depletion or accumulations occur, and it depends on the structure of the triple point. (**A**) Two grain boundaries leading to a single grain boundary will lead to mass accumulation and therefore to hillock formation. (**B**) If one grain boundary leads to two grain boundaries, there will be an area of depletion of mass causing voids.

**Figure 3 nanomaterials-09-00383-f003:**
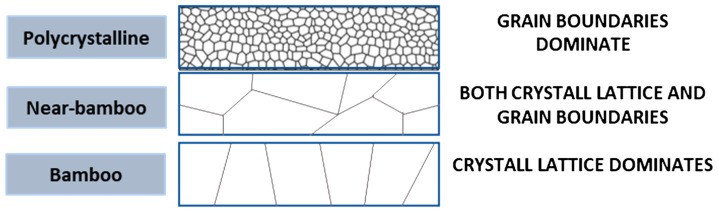
Different kinds of crystal structures which contribute to different preferred diffusion pathways during electromigration. Polycrystalline lattice has many small grain boundaries and therefore diffusion would be dominated by grain boundary diffusion although other types of diffusion also occur. On the other extreme, near-bamboo or bamboo structures will have lattice diffusion as its primary diffusion pathway, although near bamboo would also present grain boundary diffusion depending on the size of the line diameter with respect to the grain size.

**Figure 4 nanomaterials-09-00383-f004:**
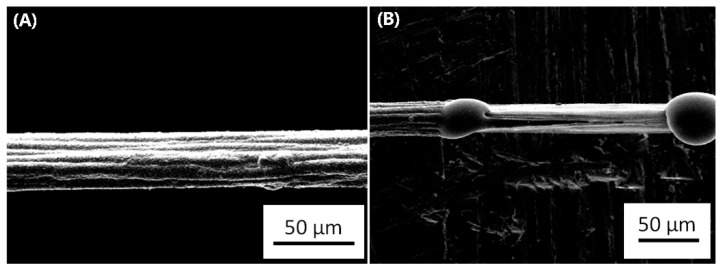
Failure due to electromigration in a graphene-copper composite. (**A**) is the line prior to applying a high current density whilst (**B**) is the same line after the failure. The image belongs to the authors.

**Figure 5 nanomaterials-09-00383-f005:**
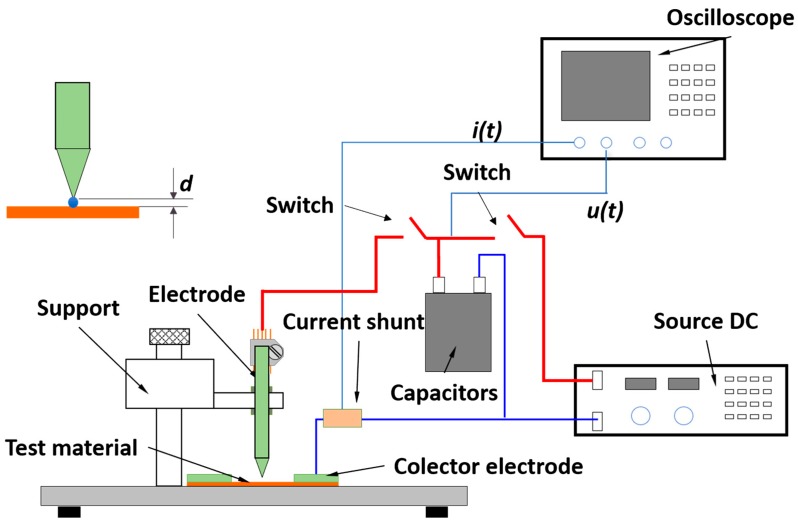
Diagram of a lightning strike system developed by the authors.

**Figure 6 nanomaterials-09-00383-f006:**
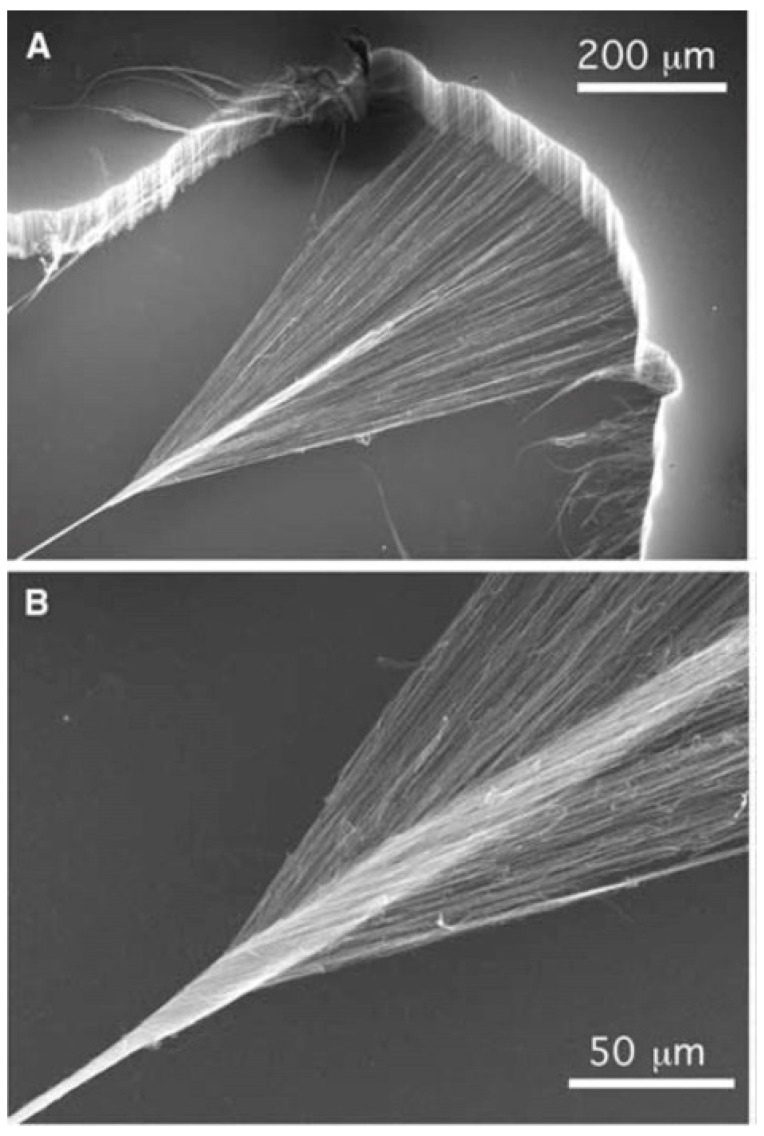
(**A**) Pulling method from a CNT vertical forest to produce CNT yarns and (**B**) CNT fibre produced by twisting of CNT yarns Courtesy of Science. Reproduced with permission from [30]. Copyright Elsevier, 2006.

**Figure 7 nanomaterials-09-00383-f007:**
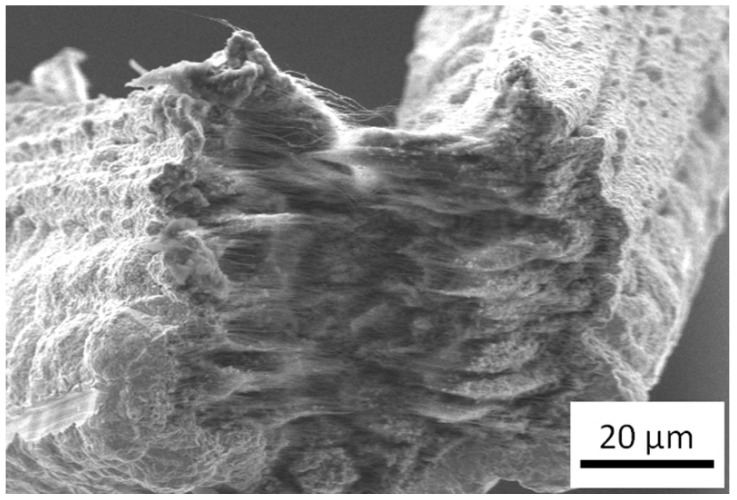
Fractured surface of CNT-Cu composite material synthesized by low current electrodeposition.

**Table 1 nanomaterials-09-00383-t001:** This table summarizes the ampacity values of several materials for comparative reasons. Two purely metallic references are placed at the beginning, to illustrate how carbon nanotube technology can improve greatly on the ampacity of these materials. Both, pure carbonaceous and composite materials exhibit ampacity properties higher than any other metal reported to date.

Type of CNT	Metal	Form of Material	Ampacity A/cm^2^	Synthesis Method	Reference Number
Non	Copper	Polycrystalline	8–18 × 10^6^	Various methods	[18]
Non	Aluminium	Polycrystalline	5 × 10^4^	Vacuum deposition	[20]
SWCNT	Non	Individual tube	1 × 10^9^	Suspension deposition	[101]
CNT	Non	Fibre	1 × 10^5^	Wet spinning method	[62]
MWCNT	Non	Individual tube	1 × 10^6^	Catalytic process	[102]
MWCNT	Non	Individual tube	1 × 10^7^	CVD deposition	[103]
MWCNT	Non	Individual tube	1 × 10^9^	Arc discharge method	[104]
MWCNT	Non	Single tube	3 × 10^6^	CVD deposition	[105]
CNT	Nickel	Test solder joint	1 × 10^4^	Ni coating on CNT	[106]
MWCNT	Nickel	Ni filled tube	1 × 10^7^	MPCVD method	[105]
MWCNT	Copper	Microscopic line	1 × 10^8^	Electroplating	[96]
SWCNT	Copper	Test line	6 × 10^8^	Electroplating	[16,107]
CNT	Copper	Fibre	1 × 10^7^	CNT aerogel + electroplating	[108]

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
