# Peer review of "High Ampacity Carbon Nanotube Materials"

_nanomaterials, 2019, doi:10.3390/nano9030383_

Round 1
Reviewer 1 Report
nanomaterials-446884: High Ampacity Carbon Nanotube materials, Guillermo Mokry, Javier Pozuelo , Juan J. Vilatela, Javier Sanz and Juan Baselga.
I read this manuscript with great interest. From the history of ampacity materials to the current issues, I acknowledge that this manuscript has been reviewed very well. Carbon nanotubes and other new carbon materials are attracting attention as high ampacity materials and nanotubes with ideal crystal structures are known to have high ampacity properties. The only thing missing from this manuscript is the relationship between the crystal structure (or defects) and properties of nanotubes. Examples of electrical and mechanical properties include those of T.W. Ebbesen et al., Electrical conductivity of individual carbon nanotubes, Nature 382, 54–56 (1996) and J.P. Salvetat et al., Elastic Modulus of Ordered and Disordered Multiwalled Carbon Nanotubes, Adv. Mater. 11, 161–165 (1999) in the past and those of Yamamoto et al, Structure-property relationships in thermally-annealed multi-walled carbon nanotubes, Carbon 66, 219–226 (2014) in recent years. Now, the crystallinity of nanotubes is known to vary greatly depending on the synthesis method, and is also known to be different by heat history. I expect that describing the relationship between crystal structure and properties will strengthen this manuscript.
Author Response
We are pleased to answer the queries of the reviewer on our paper entitled “High Ampacity Carbon Nanotube materials”. Changes in the manuscript have been written in red for an easy tracking.
REVIEWER
I read this manuscript with great interest. From the history of ampacity materials to the current issues, I acknowledge that this manuscript has been reviewed very well. Carbon nanotubes and other new carbon materials are attracting attention as high ampacity materials and nanotubes with ideal crystal structures are known to have high ampacity properties. The only thing missing from this manuscript is the relationship between the crystal structure (or defects) and properties of nanotubes. Examples of electrical and mechanical properties include those of T.W. Ebbesen et al., Electrical conductivity of individual carbon nanotubes, Nature 382, 54–56 (1996) and J.P. Salvetat et al., Elastic Modulus of Ordered and Disordered Multiwalled Carbon Nanotubes, Adv. Mater. 11, 161–165 (1999) in the past and those of Yamamoto et al, Structure-property relationships in thermally-annealed multi-walled carbon nanotubes, Carbon 66, 219–226 (2014) in recent years. Now, the crystallinity of nanotubes is known to vary greatly depending on the synthesis method, and is also known to be different by heat history. I expect that describing the relationship between crystal structure and properties will strengthen this manuscript.
ANSWER:
First and foremost, thank you very much for taking the time for reviewing the manuscript. We believe that your recommendation, indeed, could benefit the review greatly. Therefore, we have added a section about mechanical properties as you recommended and added the references you suggested. We would have liked to discuss further properties of carbon nanotubes for a longer extent, however, the space is limited and thus our attention had to be centered on one of its many properties. We hope the manuscript has been strengthened enough for your liking with this addition.
Reviewer 2 Report
The review titled “High Ampacity Carbon Nanotube materials” is focused on an unusual but at the same time very interesting problem.
The authors, in fact, discuss the ampacity which is the resistance of conducting cables to high
current densities and suggest carbon nanotubes and carbon nanotube composites as promising high ampacity materials. After an introduction section, the authors introduce the theoretical approach to the ampacity and then the experimental setup used to perform ampacity measurements.
The results, reported in the Table and taken from the literature, show that the ampacity of carbon nanotubes is able to exceed that of copper and gold. The encouraging results as well as the drawbacks when using carbon nanotubes as interconnects in electronical devices are addressed.
As previously remarked, the topic is interesting and the review is well structured, therefore I recomend the publication in Nanomaterials.
Before publication, I suggest the authors to control the word “below” at line 332 of the submitted manuscript.
Author Response
We are pleased to answer the queries of the reviewer on our paper entitled “High Ampacity Carbon Nanotube materials”. Changes in the manuscript have been written in red for an easy tracking.
REVIEWER:
The review titled “High Ampacity Carbon Nanotube materials” is focused on an unusual but at the same time very interesting problem.
The authors, in fact, discuss the ampacity which is the resistance of conducting cables to high current densities and suggest carbon nanotubes and carbon nanotube composites as promising high ampacity materials. After an introduction section, the authors introduce the theoretical approach to the ampacity and then the experimental setup used to perform ampacity measurements.
The results, reported in the Table and taken from the literature, show that the ampacity of carbon nanotubes is able to exceed that of copper and gold. The encouraging results as well as the drawbacks when using carbon nanotubes as interconnects in electronical devices are addressed.
As previously remarked, the topic is interesting and the review is well structured, therefore I recomend the publication in Nanomaterials.
Before publication, I suggest the authors to control the word “below” at line 332 of the submitted manuscript.
ANSWER:
First and foremost, thank you very much for taking the time for reviewing the manuscript, and your kind words. The word “below”, at line 332, has been changed to a better suited word. We thank you very much for recommending the publication of the manuscript.
Reviewer 3 Report
It's a good review, but the biggest challenge to the authors will be writing it effectively in English. There are many grammar errors, phrases used which come off as rough translations of colloquial phrases, some incorrect words used (line 68 with term instead of turn, for example), and even some words missing entirely (line 84: dimensions of the are, for example). This unfortunately ends up overshadowing many points the authors want to make, and is important enough to where I had this as my first point.
Example suggestions for other materials to be used instead of copper in lines 90, 91 with a reference for the mass decrease would be good
Reference needed in line 100
In line 123, do they mean mean time before failure as alluded to previously on line 119? Median time is not a quantifiable metric for a general equation, and Black's equation is generally referred to using mean time
Again for line 123, if they provide units for MTF then they should also provide units for the other variables in the equation
Degree symbol does not seem to be correctly used in line 132
Figure 1 has incorrect use of (A) and (B), since they both should be for the left column of two images showing the growth-dominated mechanism, and is missing (C) and (D) for the right column entirely
Reference needed in line 150 (lattice diffusion has the highest activation energy)
Cancelled is not the correct term to be used in line 154, replacement is more apt given the passivation layer is the new surface
No legends provided for equation 3 on line 159, and some legends missing for equation 4 on line 164
Line 165 mentions supplementary information, but I did not get it. I assume you can access it?
I am not sure if "near-bamboo" and "bamboo" are the correct terms to use in figure 3 since this is outside of my area of expertise
Reference needed for figure 4 and associated text on lines 204, 205
Lines 210-211: I would argue that Blech equation is perfectly within the scope of this review. The summary they provide just raises more questions instead.
Line 219: The authors meant to use l, but instead used 1 to use as the legend for distance along the line
Reference needed in line 232 (wind force explanation)
Should be Blech length, not Blech's length in line 262
References are not numbered in the order they are used in the text. Ref 14 is followed by ref 18 on page 7, and ref 15 is all the way down in the table on page 14. There are other such examples in the text.
Figure 5 incorrectly numbered as figure 4 again (page 8)
There should not be any room for unsubstantiated stories as legends in scientific publications (line 303)
Line 319: Iijima was not the first to observe CNTs, he just coined the term
Figure 6 incorrectly numbered as figure 5 (page 6)
Not
sure if a referee has to check whether the authors actually obtained
permission from Science to use the figure in figure 6 (currently number
figure 5). No such permissions mentioned for the other figures either.
The authors are making some very sweeping statements on the properties of CNTs compared to Cu in the first para on line 10, even though those are not necessarily true in general
Space needed between 1 and cm in line 399
Figure 7 incorrectly numbered as figure 6 (page 12)
Maybe ask them to cite our aqueous electromigration paper from Nanoscale last year in the last para in page 13?
Conclusion section is very generic, and could be strengthened with more ideas to tackle the various problems brought up over the course of the review
Author Response
We are pleased to answer the queries of the reviewer on our paper entitled “High Ampacity Carbon Nanotube materials”. Changes in the manuscript have been written in red for an easy tracking.
REVIEWER:
First and foremost, thank you very much for taking the time for reviewing the manuscript. We could clearly see that you had read the review in depth, and that is something we value greatly. We will try to answer as well as we can every single point on your review.
It's a good review, but the biggest challenge to the authors will be writing it effectively in English. There are many grammar errors, phrases used which come off as rough translations of colloquial phrases, some incorrect words used (line 68 with term instead of turn, for example), and even some words missing entirely (line 84: dimensions of the are, for example). This unfortunately ends up overshadowing many points the authors want to make, and is important enough to where I had this as my first point.
We are very sorry that you felt like the English was not good enough for this manuscript. We have done several checks of English language. We hope that this, together with all the small changes you suggested, will improve the manuscript .
Example suggestions for other materials to be used instead of copper in lines 90, 91 with a reference for the mass decrease would be good
We suggested using Carbon nanotube-copper which was the main idea behind the manuscript in first place.
Reference needed in line 100
Reference was added.
In line 123, do they mean mean time before failure as alluded to previously on line 119? Median time is not a quantifiable metric for a general equation, and Black's equation is generally referred to using mean time
You are right. This was changed to mean.
Again for line 123, if they provide units for MTF then they should also provide units for the other variables in the equation
The unit for MTF was removed. None of the variables should have had the units.
Degree symbol does not seem to be correctly used in line 132
Changed it.
Figure 1 has incorrect use of (A) and (B), since they both should be for the left column of two images showing the growth-dominated mechanism, and is missing (C) and (D) for the right column entirely
Re-drew the figure with the correct use of letters.
Reference needed in line 150 (lattice diffusion has the highest activation energy)
Reference was added as you suggested.
Cancelled is not the correct term to be used in line 154, replacement is more apt given the passivation layer is the new surface
Changed the term “cancelled” to “blocked” which we believe to be a better term for this phrase.
No legends provided for equation 3 on line 159, and some legends missing for equation 4 on line 164
Added the legend. This was a big error which we did not notice on the several read throughs.
Line 165 mentions supplementary information, but I did not get it. I assume you can access it?
The mention of supplementary information has been deleted.
I am not sure if "near-bamboo" and "bamboo" are the correct terms to use in figure 3 since this is outside of my area of expertise
There are several books and articles which use these terms. This is the reason, we used them, and we believe them to be right.
Reference needed for figure 4 and associated text on lines 204, 205
This image is ours. We have added the explanation in line 226-228.
Lines 210-211: I would argue that Blech equation is perfectly within the scope of this review. The summary they provide just raises more questions instead.
We have added a new section to talk about Blech equations. We hope this will strengthen the manuscript even further.
Line 219: The authors meant to use l, but instead used 1 to use as the legend for distance along the line
This was changed. It was a problem with the font used.
Reference needed in line 232 (wind force explanation)
Reference was added as suggested.
Should be Blech length, not Blech's length in line 262
Changed it to Blech length.
References are not numbered in the order they are used in the text. Ref 14 is followed by ref 18 on page 7, and ref 15 is all the way down in the table on page 14. There are other such examples in the text.
This was an error due to text editors. We have changed them and made sure they are in order this time.
Figure 5 incorrectly numbered as figure 4 again (page 8)
All figures changed and checked.
There should not be any room for unsubstantiated stories as legends in scientific publications (line 303)
We have removed this line entirely.
Line 319: Iijima was not the first to observe CNTs, he just coined the term
We have changed this phrase entirely. You are right.
Figure 6 incorrectly numbered as figure 5 (page 6)
Changed.
Not sure if a referee has to check whether the authors actually obtained permission from Science to use the figure in figure 6 (currently number figure 5). No such permissions mentioned for the other figures either.
All figures have been done by the Author so no permission should be required. The one which is mentioned by the referee (figure 6) is the only one which belongs to Science, and the permission was indeed granted.
The authors are making some very sweeping statements on the properties of CNTs compared to Cu in the first para on line 10, even though those are not necessarily true in general
We have checked all the facts and cross-checked them several times to make sure there were no “false” properties given for any of the materials mentioned in this review.
Space needed between 1 and cm in line 399
This was changed.
Figure 7 incorrectly numbered as figure 6 (page 12)
This was changed to the correct figure number.
Maybe ask them to cite our aqueous electromigration paper from Nanoscale last year in the last para in page 13?
We have tried using as many different articles as possible for this review, and we are sure there will be many which are unfairly left out. Aqueous electromigration was left out of this review as it was centered mostly on microelectronics, which are not usually exposed to a liquid medium where this kind of electromigration might occur.
Conclusion section is very generic, and could be strengthened with more ideas to tackle the various problems brought up over the course of the review
The conclusion section was changed.
Round 2
Reviewer 1 Report
nanomaterials-446884R2: High Ampacity Carbon Nanotube materials, Guillermo Mokry, Javier Pozuelo , Juan J. Vilatela, Javier Sanz and Juan Baselga.
With the exception of the figure numbers, necessary amendments have been made in the revised manuscript.
Reviewer 3 Report
Ok